# Microstructure and Friction Response of a Novel Eutectic Alloy Based on the Fe-C-Mn-B System

**DOI:** 10.3390/ma15249031

**Published:** 2022-12-17

**Authors:** Oleksandr Tisov, Mykhaylo Pashechko, Alina Yurchuk, Dariusz Chocyk, Jarosław Zubrzycki, Aleksandra Prus, Magda Wlazło-Ćwiklińska

**Affiliations:** 1School of Aerospace Engineering, Xi’an Jiaotong University, West Xianning Road 28, Xi’an 710049, China; 2Department of Fundamentals of Technology, Fundamentals of Technology Faculty, Lublin University of Technology, Nadbystrzycka Str. 36, 20-618 Lublin, Poland; 3Aerospace Faculty, National Aviation University, Liubomyra Guzara 1, 03058 Kyiv, Ukraine; 4Department of Applied Physics, Faculty of Mechanical Engineering, Lublin University of Technology, Nadbystrzycka Str. 36, 20-618 Lublin, Poland; 5Department of Informatization and Robotization of Production Processes, Faculty of Mechanical Engineering, Lublin University of Technology, Nadbystrzycka Str. 36, 20-618 Lublin, Poland

**Keywords:** wire-arc manufacturing, phase equilibrium, Fe-C-Mn-B eutectic system, eutectic alloy, wear-resistant alloy

## Abstract

This paper focuses on the microstructure and tribological properties of novel hardfacing alloy based on Fe-C-Mn-B doped with Ni, Cr, and Si. The 4 mm-thick coating was deposited on the AISI 1045 carbon steel by the MIG-welding method using flux-cored wires in three passes. The transition zone thickness between the weld layers was ~80 μm, and the width of the substrate-coating interface was 5–10 μm. The following coating constituents were detected: coarser elongated M_2_B borides, finer particles of Cr_7_C_3_ carbides, and an Fe-based matrix consisting of ferrite and austenite. The nanohardness of the matrix was ~5–6 GPa, carbides ~16–19 GPa, and borides 22–23 GPa. A high cooling rate during coating fabrication leads to the formation of a fine mesh of M_7_C_3_ carbides; borides grow in the direction of heat removal, from the substrate to the friction surface, while in the transition zone, carbides become coarser. The dry sliding friction tests using a tribometer in PoD configuration were carried out at contact pressure 4, 7, 10, and 15 MPa against the AISI 1045 carbon steel (water-quenched and low-tempered, 50–52 HRC). The leading wear phenomenon at 4 and 7 MPa is fatigue, and at 10 and 15 MPa it is oxidation and delamination.

## 1. Introduction

Eutectic materials and coatings are vital for modern industrial manufacturing and reconditioning [1,2,3,4,5]. Fe is typically used as a matrix for such applications [1,2,3]; Co and Ni are used more seldomly due to their high cost. Fe and Fe-based materials for friction applications are essential [6]. They are known to be inexpensive but very efficient as structural and functional materials and as tools [7]. Carbides, borides, and other strengthening phases may be utilized to increase their strength, wear resistance, and durability. The particles may be formed in situ (eutectic and eutectoid alloys), mixed with alloyed or elemental powders, and then sintered. In both cases, the resultant metal consists of a matrix (Fe-based solid solution) and hard particles are dispersed [8]. Cr may enhance the metallic phase, Ni, W, and other elements to change the phase transition dynamics and to produce ferrite, austenite, or martensite (after quenching or air-cooling). Depending on the quantity and type of second-phase forming elements, the strengthening particles may be laths (pearlite), fine particles (honeycomb or rosette crystals), and other shapes. However, typically, the alloy contains hard particles of only one type (for example, only carbides or intermetallic precipitates). Depending on the content of, say, carbide-forming elements and C, carbides may be different in size, shape, and stoichiometry, thus providing the alloy with different properties. In works [9,10], carbide-forming elements are added to refine the boride microstructure. A total of 1–14% of V increases wear resistance by reducing the boride grain and forming minor amounts of VC carbide, changing the mode of preferential carbide growth to staggered [9]. The drawback of this is the significant content of V in the material, which makes it quite expensive. Adding V in the duplex Fe-B alloy in even higher quantities is also beneficial [10]. Other additives, such as Ti and Nb, also improve the wear resistance of the eutectic Fe-C-Cr alloys [11,12]. Mo in the Fe-B alloys does not form carbides, but replacing the Fe in M_2_B _2_B increases its toughness and abrasion resistance, with a minor effect on boride grain size [13]. The addition of tungsten does not affect the size, but amounts of 1–3% improve the toughness and wear resistance of the alloy [13]. WC particles may be added to F-Cr-B alloys to improve wear resistance by increasing the content of carbides in the alloy up to 85–88% [14].

In contrast, the ex-situ introduction of titanium carbides reduces the alloy’s wear resistance obtained by fusion deposition [12]. Coarse and hard TiC particles easily separate from the parent material and damage the friction surface. Pre-prepared carbides may be effective but also make the material more expensive.

The balance of matrix and filler properties provides the eutectic material’s wear resistance. However, too big a difference creates additional stresses in the matrix, and carbides spall off the surface [15]. Previously, in many steel tools, cementite was the essential strengthening phase [16,17,18]. Its morphology and properties can be changed via heat treatment and alloying over a broad range [19,20,21,22]. Later, it was replaced by Cr_x_C_y_ carbides due to their higher hardness; their properties may be modified over a broader range [23,24,25].

Hardfacing coatings containing both carbides and borides have been known for decades [25]. They are easy to deposit by most methods, cheap, and highly available. However, their properties may be improved. Small additions (2%) of Mn significantly improve the ductility of M_2_B borides, while the microstructure remains nearly unchanged. The other alloy with higher Mn content was studied in ref. [26]. It also had no effect on the microstructure, but it was beneficial for wear resistance. Mn partitioning in the Fe-C-Mn-Si alloys improves carbide formation and may replace Fe in cementite [21,27]. Mn is a well-known austenite stabilizer, and it also improves the corrosion resistance of steel [28,29].

Si in Fe-based alloys in quantities of 3–6.5% improves magnetic properties [30,31], oxidation resistance [31], and mechanical properties [30,31]. In alloys containing Mn, it promotes the formation of a compound oxide layer of (Fe, Mn)O and (Fe, Mn)_2_SiO_4_ and arrests the hydrogen permeation even more effectively than Al [32]. In this way, it increases the stability of the alloy in H-charging environments. In high-Mn FeMnAlMo steels, 1% Si twice increases the precipitation of carbides and significantly improves its deformation behavior [33,34]. The other austenite stabilizer, Ni, is also often added to steel and wear-resistant alloys. Even in microstructures, it can significantly improve the properties of Fe alloys after quenching, the same as Cr [35]. Increasing Ni content from 1.5 to 3% in 30CrMnSiNi2A HIP-manufactured alloy steel significantly increases its strength and toughness and improves fracture resistance [36]. A continuous increase in Ni leads to improved ductility, toughness, and impact resistance in Fe-based alloys [37]. Together with Cr, it improves oxidation and corrosion resistance [38,39,40,41]. Fe-C-Mn-B alloys may be effectively used as a coating material.

The Fe-Cr-C-B eutectic alloys are well-known to have high wear resistance and good mechanical properties [25,42]. Their bending strength may be increased by nanomodification, which significantly decreases the second-phase particle size and makes their distribution more uniform [43]. The Ni-Cr-B-Si-Fe coating was tested at room and high temperatures. Even at 700 °C, it was effective [44]. In ref. [45], the basic Fe-C-B-Cr system was modified with W. It is strengthened with carbides and borides synthesized in situ. The vein-shaped carbide structures provide an interlocking effect and improve wear resistance. The Ti-modified coating was studied in ref. [46]. The increasing content of B above 0.99% in Fe–15Cr–2.5Ti–2C–xB hardfacing eutectic alloy leads to an increase in carbides and wear resistance. The alloys based on the Fe-C-Cr-B system also have good abrasion resistance [47,48].

The above-reviewed Fe-based materials have 3–4 primary alloying elements, which provide the required properties. Ni, Cr, and Mn are of primary importance. In addition, Si may impart oxidation resistance, which is especially important for coatings deposited by fusion methods. The shield gas environment protects the surface inside the flame torch or around the electric arc. However, during coating, the deposition spot moves, and the protective gas also moves, but the solidified coating surface remains hot and actively oxidizes. Here, the function of Si is to suppress this process. From this perspective, we decided to add Si to a new eutectic material to be used in the wire-arc and spray-deposition methods.

In this study, we combined the properties of carbides and borides into a single material in substantial quantities. The M_2_B borides are more wear-resistant than carbides but are also harder and create more stress in the matrix. Based on our earlier published research [49], we decided to improve the performance of the basic Fe-C-Mn-B alloy with Cr, Mn, and Ni for solution strengthening and the (Cr) formation of carbides, and Si was used to reduce oxidation during the coating deposition process. The researched material is to be used as a coating to recondition the worn surfaces of car tire shredder blades. These blades operate in dry conditions, and mainly cut three materials: rubber, textile fibers, and steel cord wires. The latter is the hardest tire component and is made of medium or high (0.6–0.9%) carbon steel in tempered conditions [50,51,52,53,54]. From this perspective, as a counterpart, the quenched and tempered steel AISI 1045 was selected due to the ease of heat treatment, high hardness after quenching, and availability in the required size.

## 2. Experiment

### 2.1. Materials

To produce the coating via the flux-cored wire, it was necessary to balance the composition of the filling powder (Table 1) to obtain the target alloy after fusion with the steel of the envelope. The powder mixture is an initially produced [49] eutectic alloy doped with Si, Ni, and Cr. Si increases the oxidation resistance and thus improves the weld surface quality, reducing the likelihood of hot cracks between the welds. Ni is a well-known improver of steel ductility and toughness. Additionally, it increases corrosion and oxidation resistance. The addition of Ni also improves the strength of the solid Fe solution and stabilizes the austenite at room temperature. Here, Cr, together with Fe, is the basic carbide former. It also improves oxidation and corrosion resistance, hardness, and solid solution strength. Therefore, mixed Fe-Cr carbides and borides are used in the chemical composition of this coating.

The argon-atomized powder had an average particle size of 150–200 µm (Figure 1a). For the deposition, a powder-cored wire was fabricated at Wolco Sp. Z.o.o., Lublin, Poland. We used the 13 × 0.35 mm strip made of 06JA low-carbon steel as the envelope. It was wrapped into a Ø4 mm tube with no overlapping. The wire volume-fill ratio was 0.35. Furthermore, the flux-cored wire was cold-rolled until its diameter was reduced to 2.4 mm. Coating deposition was performed using the Evolution Pro 5200 (Kemppi, Lahti, Finland) equipment at the deposition current of 260–280 A and voltage of 30 V with argon-gas shielding. The 4 mm-thick coatings were obtained in 3 weld passes; see Figure 1b. The sample consisted of a substrate material (tempered medium carbon steel AISI 1045), a coating boundary (5–10 μm), and a place where two materials were fused and mixed. There are also two weld transitions (~80 μm) between the subsequent welds, in which the microstructure differs from the central weld area.

### 2.2. Research Methodology

Tribological tests were carried out in laboratory conditions using a PoD tribometer THT-800 (Anton Paar, Buchs, Switzerland), where the AISI 1045 steel disc (Ø60 mm, 10 mm in thickness) was used as a counterpart. It was quenched in oil at a temperature of 800–820 °C and tempered at 350 °C. The hardness of the counterpart material was 50–52 HRC ±4HRC. The samples had a rectangular 10 × 10 mm shape. Both disc and pin were polished with a SiC suspension to the roughness Ra = 0.1 μm. The dry sliding friction test was performed at a contact pressure of 4, 7, 10, and 15 MPa, and the friction path was 8640 m at a speed of 0.4 m/s, with a test duration of 6 h [7,8,20].

The tested samples were cut along the direction of the friction, and the surface and cross-sections were studied. Metallographic samples were made using the typical methodology: Mecatech 234 automatic polisher (Measurement and Control Solutions Corp, Miami, FL, USA) with water cooling at 150 rpm. Samples and counterparts were ground with SiC sandpaper (the finest was 4000 grit) and finally polished with a 0.05 μm polycrystalline diamond suspension. After each preparation stage, the samples were cleaned in a sonicator for 30 s to avoid surface scratching by the retained abrasives. The cross-section surface was not etched to prevent any chemical changes to the coating. For the general topography observations and estimations, we used a laser confocal microscope, VK9710K (Keyence, Singapore). The micromechanical characteristics of coating components were determined using the Hysitron T-950i triboindenter (Bruker, Billerica, MA, USA), with a Berkovich indenter at a load of 5 mN. We used a SU3500 microscope (Shimadzu, equipped with the Oxford EDS system (Oxford Instruments, Abingdon, UK) for the SEM studies. The secondary ion mass spectroscopy was conducted using the IMS-4F instrument (Cameca, Paris, France). The D8 Advance diffractometer (Bruker, Billerica, MA, USA) acquired the XRD-spectra in Bragg–Brentano configuration (2θ = 10–90°, scanning speed 10°/min), followed by the phase identification of the friction surface and intact material with HighScore Plus software (version 3.0e, Malvern Panalitical, Malvern, UK) using the Crystallography Open Database.

## 3. Results and Discussion

### 3.1. Phase Identification

After preliminary microanalyses, the alloy’s microstructure was determined as an Fe-based matrix with two types of strengthening particles: light gray and dark gray. The high content of C and B in the alloy should result in both borides and carbides. However, the EDS analyses are not highly sensitive to both C and B. To associate black and gray particles with borides or carbides, we cut the weld portions and remelted them in the vacuum furnace. One of them was furnace-cooled, and the second was water-cooled. Due to the different diffusion rates of C and B in Fe, this trick should result in different amounts of borides and carbides (dark gray and light gray particles). XRD analyses may trace this change. We also assume that, as B is lighter than C, borides on BSD images (chemical mode) should be darker [5].

Both samples have a casting microporosity (black arrows). Increasing the cooling rate results in a slight increase in the pores’ quantity and size. The fast cooling (Figure 2a) results in the uniform composition of strengthening particles, although they vary significantly in size. The biggest particle passes through the image and is approximately 20 µm wide. The smaller particles are near-equiaxed (10–15 µm) and elongated (2–5 × 30–40 µm). After slow cooling (Figure 2b), two types of particles appear: dark gray (near-equiaxed and elongated, red arrows) and light gray. In the red rectangle, one may see that some darker grains grow from inside the lighter ones.

The XRD spectra acquired from the surface are given in Figure 3. The analysis of the diffraction patterns reveals that Cr_7_C_3_ and M_2_B _2_B phases appeared in both samples [13,49,55,56]. The positions of the peaks in the X-ray diffraction profiles correspond to the following phases: Figure 3a: γ-Fe (11.9%), α-Fe (7.9%), M_2_B _2_B (3%), and Cr_7_C_3_ (59.4%); Figure 3b: γ-Fe (5%), M_2_B (79%), and Cr_7_C_3_ (15%). There are also some unidentified phases. This means that slow cooling produces both Cr_7_C_3_ carbides and M_2_B _2_B borides, while fast cooling results mainly in Cr_7_C_3_ carbides on the surface. Thus, we may firmly associate darker grains with M_2_B particles and lighter ones with Cr_7_C_3_.

### 3.2. Microstructure Observations

The microstructure of the coating is shown in Figure 4. The substrate’s microstructure is not resolved. However, one may see that the thickness of the boundary layer is 5–10 µm, and it is a solid solution with no visible second-phase inclusions. Above it, the first layer of the weld consists of an Fe-based solid solution with fine, light carbides (according to Section 3.1) with only a few darker borides in the left-upper corner. The EDS mapping of this coating (Figure 4c) clearly indicates the boundary between the coating and the substrate, as noted in [32].

Above the boundary line, two types of carbides may be resolved. In the left part of Figure 4a, they are separate, coarser particles, being 5–10 µm long and 2–3 µm wide. In the central part, they are united in rosette structures and are much finer, being 1–5 µm in length and approximately 1 µm in width. As the substrate material before coating deposition was not preheated, the cooling rate here was high, preventing the formation of borides.

Figure 4b shows a high-magnification image of the substrate-coating interface. The chemical composition of the analyzed points is given in Table 1. The waved line between the boundary and weld 1 (the structure of Figure 4b is the same as for Figure 4a) indicates the mixing of the coating and substrate materials, which should provide good adhesion strength (this was not studied here). The carbides, Figure 2b, points 5 and 6, stretch along the boundary, the same as the carbides around point 4. They indicate a threshold beyond which carbides do not form or cannot be resolved. The particles, as in points 1 and 2, are similar to those in Figure 4a, the central part. According to qualitative EDS analyses (see Table 2), they contain more C (22–24%) than boundary carbides 5 and 6 (19–20%).

The area of solid solution in the weld layer (pp. 3 and 4) is a fast-formed austenite containing alloying elements, listed in Table 1. B was not detected in this area. The composition of the solid solution in points 7 and 8 is different, containing a lower amount of Mn, Ni, and Cr when compared with points 3 and 4 but a higher content of Si. This difference suggests that the coating and substrate materials are mixed and form a boundary. The boundary composition is closer to the nominal coating than the substrate. The increased Si content here (Table 2, points 7 and 8, Figure 4c, arrowed) may be explained by its fluxing action during the welding process.

Figure 5a displays the transition zone between weld layers 1 and 2. The red line separates the sample’s two different microstructures. Below the line is a transition layer formed as a result of mixing two subsequent welds. Increasing the temperature (e.g., by tempering) of the lower layer during the deposition of the upper layer leads to the growth of carbides and bimodal borides (dark, longer grains).

The 20–80 µm-long and 5 µm-wide borides are mainly arranged along the direction of heat removal (see Figure 5a). Small borides (5–40 µm long and 1–3 µm wide) are randomly oriented. We may assume that bigger borides were initially nucleated during metal solidification (long grain in Figure 2a) and grew bigger during the second layer deposition, and the smaller borides appeared due to reheating. The temperature in the transition layer was high for a more extended period, as the borides were shielded by the second weld.

The central area of the weld (Figure 5d) consists of rosette or feather-type carbide crystals with few boride particles in the right-bottom corner. In the center and left-upper corner are the 5–15 µm areas of solid solution. This microstructure correlates with the conclusion drawn in Section 3.1 and represents a fast-cooled material consisting mainly of austenite and Cr_7_C_3_ particles with a minor amount of M_2_B.

The studied eutectic alloy with the composition of Fe-0.24C-5.1Mn-1.36B-4.4Si-5Cr-4.8Ni is composed of an austenitic solid solution with a minor content of ferrite, which may form at high cooling rates. As a strengthening phase, both borides and carbides appear. The varying cooling rate is effective in controlling their size. The Cr_7_C_3_ carbides are present in the form of 1–3 μm particles in the transition zones and rosette arrays inside the single layer. M_2_B may form long crystals directed toward heat removal. Smaller borides are randomly dispersed inside the coating. The production of the coating by Ø2.4 mm flux-cored wire made of a steel 06JA sheath filled with argon-sprayed powder results in a dense and well-bonded junction with the substrate.

### 3.3. AFM Observations and Nanohardness Tests

The nanohardness test was used to determine the nanohardness of the coating components. The results are presented in Figure 6 and Table 3. The measurements were taken in the middle of the sample, near the transition zone between weld 2 and weld 3. As mentioned above, the alloy consists of three main constituents: Fe-based solid solution (points 3, 4, 8; Figure 6a–c), carbides (points 1, 2, 7; Figure 6a,c), and borides (points 5, 7; Figure 6b,c). The hardness of borides is superior to the hardness of the matrix and carbides. This value is the same (see Table 3) for all transition zones. For example, the hardness of small carbides in the transition zone (Figure 6c, point 7) is 16 GPa, which is 30% less than the hardness of borides in the same area, but in the center of the weld layer, they are harder, being 19–20 GPa. 

The hardness of the metal matrix is 3.8 GPa between small carbides (point 3), 6.17 between the borides, and 4.58 GPa between small carbides and long boride particles (point 8). This scattering may be explained by the fact that the distance between the indenter tip and the particles is not considerable, and may restrict the penetration, thus virtually increasing the hardness. Hence, the hardness of borides was determined accurately, while for the matrix material, it is in the range of 3.8–6.57 GPa. Based on the data from Table 3, the average measured value of the hardness for borides is 23.4 GPa, for carbides is 17.7 GPa, and for the matrix is 4.85 GPa.

The hardness profile of the coating subfriction area of the sample, tested at the contact pressure of 15 MPa, is shown in Figure 7. Here, only the matrix material was tested. It was built in order to trace the friction-induced strain in the metallic component of the weld. From Figure 7, it can be seen that the thickness of the friction-affected layer is 8 µm (first three indents). Indent 4 (5.13 GPa) has a value close to the one determined earlier for the matrix material (see Table 3, indents 3, 4, and 8).

The first indent was made at a depth of 4 µm, and the hardness value is 12.3 GPa, which is more than twice the average value. This value rapidly drops and reaches only 5.13–5.15 GPa at a depth of 10–22 µm. Then, it rises to 6.77 GPa at a depth of 85 µm and drops again. We may assume that the depth of 120–140 µm is the limit, where the friction force effects are localized because the matrix hardness changes insignificantly below this level.

The hardness profile indicates that the major friction-induced changes are in the top 8–10 µm, and this layer is responsible for the wear process.

### 3.4. Coefficient of Friction and Wear Loss

The change in the coefficient of friction is shown in Figure 8, and its values are given in Table 4. The dark lines show the smoothed CoF values, and the light lines show the actual measured values. For each graph, three sections may be identified. The first section has an initial plateau where the CoF is at its maximum. In the second, the friction coefficient decreases to a low value and then becomes steady. In the third section, it fluctuates around some equilibrium value (for the current conditions). Nevertheless, the character of the decrease is dissimilar for different contact pressures.

For the sample tested at 4 MPa, the plateau is the longest and lasts for 5000 s, and for the sample tested at 15 MPa, it is only 3000 s. So, the running-in period decreases as the contact pressure increases. The transition from running-in to steady friction also decreases. The most significant difference between the steady and running-in values is for the sample tested at 4 MPa: more than twice. The lowest difference is for the sample tested at 15 MPa: only 32%. Considering the nature of CoF curves, it may be said that friction is steady for all considered contact pressures. The catastrophic surface destruction was not detected.

The wear loss of the tested samples is given in Table 4. The increase in contact pressure from 4 to 7 MPa results in increases in wear by order of 1.9; from 7 to 10 MPa, by 1.7; and from 10 to 15 MPa, by 2.3. Here, the increment in wear loss is not linear, and the biggest value is obtained for the contact pressure change from 10 MPa to 15 MPa. The increase in contact pressure in the order of 3.75 (from 4 to 15 MPa) causes a 7.4 times increase in wear loss.

### 3.5. Surface Characterization

#### 3.5.1. Cross-Section Analyses

The sample cross-section tested at 15 MPa is given in Figure 9. In Figure 9a, the long boride grain has several horizontal cracks in the upper portion shifted in the friction direction. It consists of the matrix, carbides, and one long boride particle. The boride in the center of Figure 9a is deformed towards the action of friction. The depth of cracks is within 10 µm. Similar behavior was noticed in refs. [9,21]. This means that the major friction-induced stresses are localized here. This is also proved by the increased strain described in Section 3.3, Figure 7. The red line is on the same level as the surface of the boride, M_2_B _2_B. It is approximately 1 µm lower than the sample surface behind it (if it moves in the friction direction). We assume that a cracked-off micron-sized fragment of the boride may result in additional surface damage during friction, as it is harder than the other alloy constituents. For future studies, it is therefore better to avoid the formation of vertically oriented borides or to make them refined, as in Figure 9b.

In Figure 9b, the surface layer contains only small borides and carbides, which are coarser than those in Figure 9a. In Figure 9, the surface layer is integral for both micrographs, with no evidence of strengthening particle spallation. They are well-joined to the matrix, and cracks between the matrix and the particles were not detected. Generally, the microstructures below the wear scar are very similar to those of the intact one; see Figure 4 and Figure 5.

#### 3.5.2. Surface Roughness

Figure 10 shows the topography and roughness measurements of the coating tested at 4, 7, 10, and 15 MPa. These images were created using the laser confocal microscopy method, and the surface fragment is accompanied by its 3D map. Red arrows show the friction direction.

In Figure 10a,c, there are diagonal polishing scratches. The surface is only slightly damaged and has near-circular dark sites. If referring to the material’s initial microstructure, they may be identified as rosette clusters of carbides, and primary friction plateaus emerge over them. The wear debris starts to accumulate on the friction surface. The general friction roughness is only twice that of the initial (0.1 μm vs. 0.22 μm). The boride particles were not identified.

When the contact pressure increases, the roughness of the friction surface also increases. At 7 MPa, Ra = 0.29—a 24% increase. The increase in roughness here is less than the increase in CoF and weight loss, which may be explained by the removal of fine particles from the surface. Indeed, worn particles on the surface (Figure 11b,d) are 0.5–5 μm in size. At 7 MPa, the secondary plateaus of friction contact appear, built of wear debris. The diagonal polishing lines are still clearly visible.

At 10 MPa, the friction surface is covered with oxides, and despite a higher coefficient of friction, it is smooth, and the wear process is stable, with no severe surface destruction detected. The surface roughness rises to Ra = 0.37 μm, and the mean depth of pits and cavities is within 0.37 μm. The wear process occurs via general surface abrasion, the delamination of the friction-induced structure, and the formation of spall pits.

The most damaged surface is that of the sample tested at 15 MPa. The average depth of pits is greater than 6 μm, and the surface roughness Ra = 0.56 μm, which is 5.6 times bigger than the initial value. The plateaus of the secondary friction-induced structures cover only half of the surface. They delaminate (Figure 10g) and crack off at the edges (Figure 11b), and, because of this, cannot entirely cover the friction surface. As a result, the newly formed alloy area is directly subjected to friction. 

Figure 11 presents the topographic SEM images of the friction surface. Red arrows show the friction direction. The sample tested at 4 MPa is shown in Figure 11a. The black structure along the friction direction is formed from the wear debris. Figure 11b shows that the light particles, 1–2 μm in size, are embedded into a darker substance. We may assume that they are the carbides spalled off the surface, as they are close in size and shape.

The process of carbide spallation may be seen in Figure 10a,c. Additionally, the apexes of vertically oriented boride particles are shown in Figure 11b. Despite their position and orientation, they are covered by the friction-induced layer, contributing little to the destruction of the friction partner. 

The fragments of these borides (see Figure 9a) are mostly flat and thin and are not detected in the structure of the secondary plateaus. The thickness of the compacted mixture of wear debris is approximately 10–15 μm, as may be estimated from Figure 10b, where visible signs of delamination can be seen. On the sides, cracks and fragments may be refined and compacted again, or removed from the friction area.

The surface of the sample tested at 7 MPa is not covered by a protective friction-induced structure (Figure 11c). The diagonal array is a typical surface destruction site, where the carbides bear the friction stress and start to spall off the surface. Black spots inside the outlined region are the voids: places where loosed carbides are found in the matrix. Around the area, the carbides are still firmly joined with the metal. 

Thin oxides formed on the surface at lower loads (4 and 7 MPa), and, after being worn off, accumulate between the polishing and friction grooves (Figure 10a,c and Figure 11a), as may be seen in Figure 11. The dark spots and lines (scan points 4–6, Figure 11b, Table 5) are in the valleys of the wear track and are covered with the oxidized wear debris. The oxygen content here is much lower than that found in other studies [4,8,26,42,43]. Wear debris cannot be compacted in these conditions and only eventually form secondary contact plateaus (Figure 10a,c).

Furthermore, fragments of carbides and borides are produced, which are mixed with other worn particles at higher loads (10–15 MPa) to form the secondary friction-induced layer [26,57]. It can be seen that despite being unprotected by oxides, the alloy, coupled with hardened steel AISI 1045, has high wear resistance in dry sliding. On the one hand, the high overall content of Mn, Cr, and Ni makes it oxidation-resistant, and on the other hand, the formed thin oxide layer is stable and effective in separating the friction surfaces. Referring to Figure 9, Figure 10 and Figure 11, the adhesion wear mechanism cannot be detected, and the coefficient of friction (Figure 8) is typical for steels (0.2–0.3, [58,59]), being much lower than 0.35–0.58 in [60].

The scanned points 1–3 (Figure 12) have low oxygen content, much lower than for friction-induced secondary structures. The EDS mapping (Figure 12c) also indicates low oxygen content. We assume that, at 4 and 7 MPa, the heating of the friction surface is not sufficiently high to promote oxidation, while at 10 and 15 MPa the surface is covered by the oxide layer. A high load and intensified oxidation promote the higher wear loss of the sample.

As may be seen from Figure 9, the structure of the alloy near the friction surface is the same as inside the sample. Thus, friction did not create a sufficient temperature to cause the carbide/boride grain growth. As was shown in our earlier study, the Fe-C-Mn-B system is typically stable and demonstrates no phase transformation up to 1000 °C. The average surface temperature measured by contactless laser pyrometer after 5 h of testing was 310 °C for 15 MPa, 225 °C for 10 MPa, 182 °C for 7 MPa, and 158 °C for 4 MPa.

#### 3.5.3. Analyses of Friction Surface SIMS Patterns

The next stage was using SIMS secondary ion mass spectrometry for the quantitative analysis of chemical elements. The acquired spectra are given in Figure 13. The surfaces were subjected to ion etching. The quantitative analysis of the characteristic friction areas at a depth of 5–10 nm is given in Table 6. The surfaces of the samples (eutectic alloy coating) and counter-samples (steel AISI 1045) were examined before and after friction.

If we compare the data from Table 6 with Table 1, we can see an increased content of alloying elements (especially Ni and Cr) in the thin surface layer. During friction, the Cr and Ni diffuse to the surface and form surface oxides. With the increase in the contact pressure and, thus, the temperature at the same time, the content of light elements decreases by nearly twice for C and 5–6 times for B. The maximum B content in the 5–10 nm surface layer is found for 7 MPa, and the lowest is in the intact surface. The content of Fe varies in the range of 58–63%, with no correlation with contact pressure, the same as for C and Si. Si is the only element for which the content for all conditions equals the nominal value. 

The content of Ni and Cr depends on contact pressure: as the load rises, the Ni content decreases, and the Cr content increases. The content of O is also the highest for 10 and 15 MPa. Additionally, the oxygen content on the surface of the sample, tested at 7 MPa, as determined by EDS and SIMS, is very close.

Taking into account the content of alloying elements in the tested material and their distribution in the thin (5–10 nm) surface layer, we can say that Ni and Cr are primary elements that take part in the formation of the protective oxide layer, which separates the friction partners and reduces the coefficient of friction at 4 and 7 Mpa. 

The thin layer of oxides effectively separates the sample material and the friction partner, despite the high content of hard phases. In a 5–10 nm thin surface layer, the content of Cr, Ni, and Mn is increased and is much higher than the nominal value. This may be caused by an increased surface temperature and the easy diffusion of these elements in the Fe-based solid solution, while the diffusion of interstitial elements C and B is slower, and their content is lower. On the other hand, the content of Si is close to the nominal value. 

At 4 and 7 Mpa, the mean coefficient of friction in the steady period is 0.2 and 0.28, respectively. Here, the wear process is mainly due to fatigue-caused carbide spallation (Figure 9a,c). The actual contact areas arise in the sites where the matrix is strengthened with a carbide Cr_7_C_3_ mesh. The matrix material is much more ductile and wears off easier. In the friction interface, its particles oxidize and accumulate in the friction grooves (Figure 10a), surface valleys, and spall pits. 

The carbides and borides bear a major part of the friction load, as the coating surface is significantly damaged around them. Due to cyclic loading, they spall off, leaving spall pits, as shown in Figure 10c (black spots in the outlined region). This process is clearly seen at the contact pressure of 4 and 7 Mpa, as the wear site is not covered with oxides. 

The CoF Is higher at 10 and 15 mPa, being 0.39 and 0.57, respectively. The friction-induced layer comprises worn oxide particles and spalled carbide and boride particles. These particles are slightly fragmented, mixed with worn oxide particles, and accumulate on the friction surface, thus forming the secondary plateaus of friction contact (Figure 10b). 

The surface roughness also increases if compared with the samples tested at 7 and 10 MPa (see Figure 10). The increased stress and oxidation promote a more intensive wear process (see Table 4). Here, at 10 MPa and with a steady period of friction, the wear occurs on the secondary plateaus, which almost entirely cover the surface (Figure 10e). The main damage occurs due to abrasion, delamination, and side cracking. With the increase in contact pressure, the amount of damage to friction-induced layers increases, and even juvenile metal areas appear. Thus, the wear loss continues to grow.

## 4. Conclusions

In this research, the tribological behavior of the Fe-0.24C-5.1Mn-1.36B-4.4Si-5Cr-4.8Ni MIG-welded coating obtained in three-electrode passes was studied. The powder mixture was specially balanced to obtain the target material using flux-cored wires. In the dry sliding pin-on-disc tests, the material exhibited high wear resistance and suitability for long-term service. Based on our results, we may draw the following conclusions: The strengthening of alloy with both Cr_7_C_3_ carbides and M_2_B borides is effective in providing the coating with high wear resistance, via mechanical interlocking between the matrix and strengthening particles. Hard borides that are normally oriented toward the friction surface (22–23.4 GPa) with the mesh of carbides between them (16–19 GPa) are bonded with a ductile and relatively soft matrix (5–6 GPa). Consequently, they bear a major part of friction stress, especially at loads of 4 and 7 MPa, and provide a steady coefficient of friction of 0.2 and 0.28, respectively. At 10 and 15 MPa, the steady friction coefficients are 0.39 and 0.57, respectively.The friction process activates the diffusion of alloying elements. Ni, Cr, and Mn move towards the surface, and their content in the 5–10 nm thin surface layer is much higher than nominal. As the contact pressure (and contact temperature) increase, Cr content rises to 18% at 15 MPa (11.68% in the intact surface and 5% nominal). The content of Ni is at its maximum on the intact surface and reduces as contact pressure rises (16.47% and 13.32%; nominal content is 4.8%). The Mn content is 1.6–1.9 higher than the nominal value (5.1%), and at 15 MPa, its content is only 3%. Thus, there is no direct relationship between the contact pressure and Mn content in the friction-induced surface. The content of C and B is lower than the nominal value, while the content of Si remains almost unchanged during all friction tests.There is a transition between different wear mechanisms. At a contact pressure of 4 and 7 MPa, there is the fatigue spallation of carbides, while at 10 and 15 MPa, there is oxidation wear accompanied by delamination.

## Figures and Tables

**Figure 1 materials-15-09031-f001:**
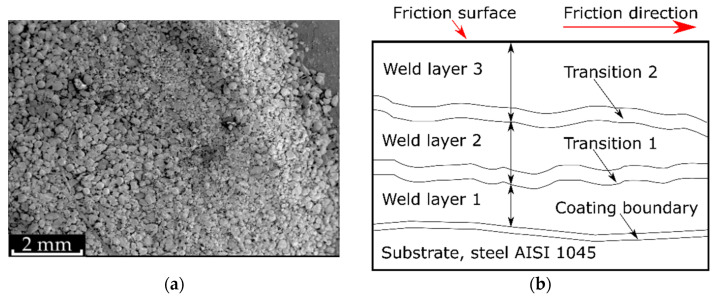
The structure of the studied sample: (**a**) the morphology of the filling powder mixture, and (**b**) the layout of the coating deposited be MIG-welding in three weld passes.

**Figure 2 materials-15-09031-f002:**
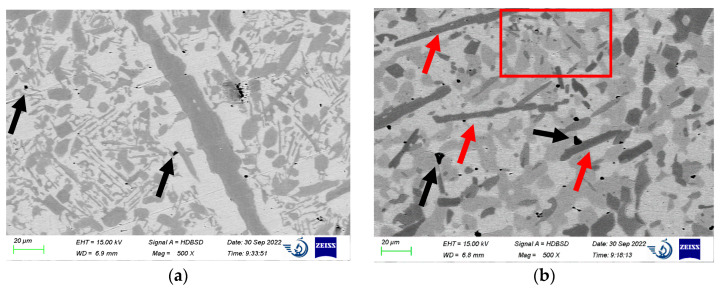
Microstructures of water-cooled (**a**) and furnace-cooled (**b**) samples.

**Figure 3 materials-15-09031-f003:**
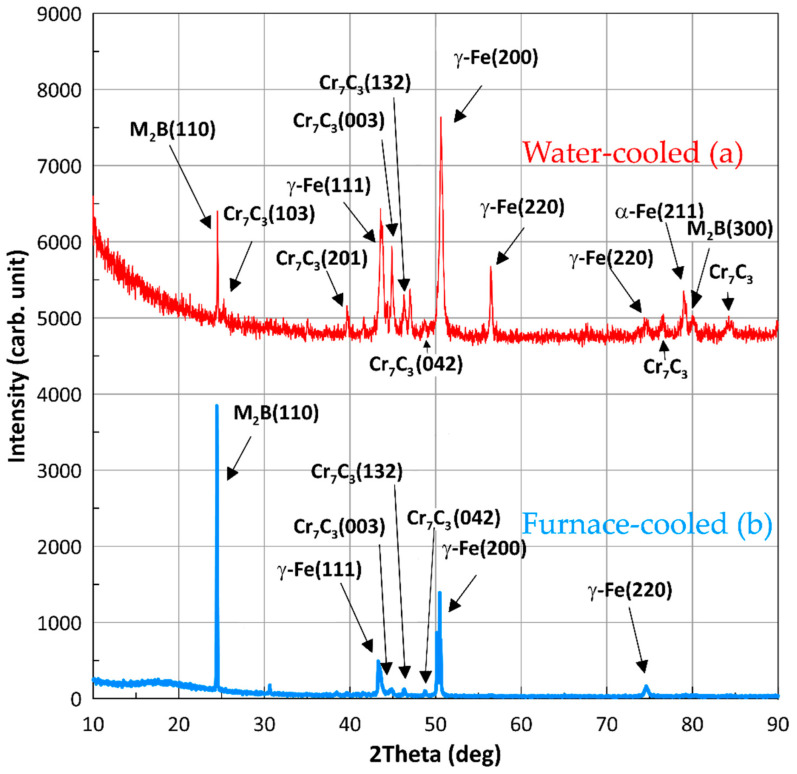
X-ray diffraction patterns for the Fe-C-Mn-B-Cr-Ni-Si alloy obtained under water (**a**) and furnace cooling (**b**).

**Figure 4 materials-15-09031-f004:**
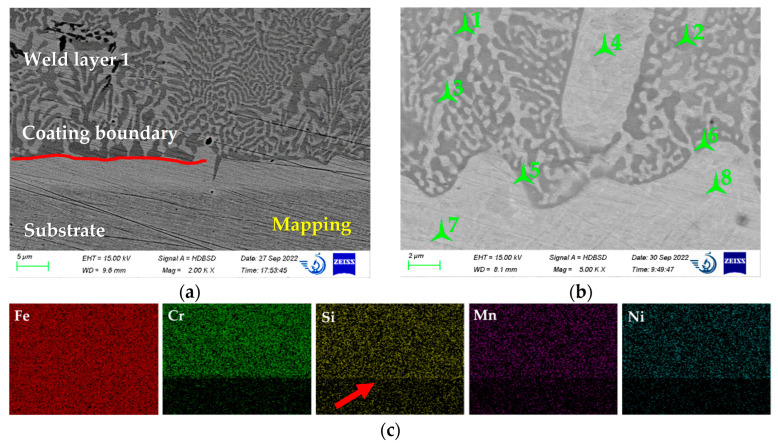
The structure of the obtained coating and EDS mapping of the substrate–coating interface: (**a**) general layout, (**b**) points location for the EDS point analyses, and (**c**) EDS maps of the coating–substrate interface.

**Figure 5 materials-15-09031-f005:**
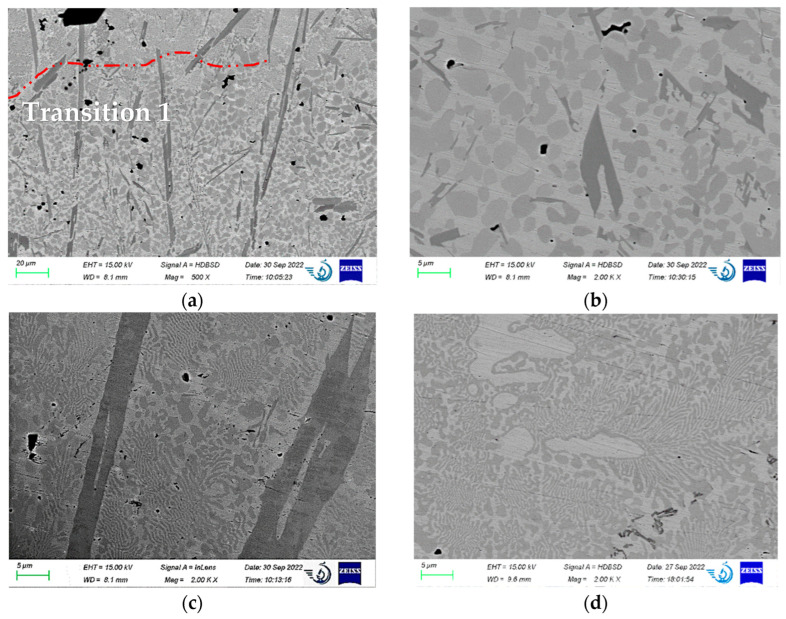
Microstructure of the coating: (**a**) transition layer 1–weld 2 interface, (**b**) microstructure of the transition zone (below the red line in (**a**)), (**c**) microstructure of the weld just above the transition layer (above the red line in Figure 3a), and (**d**) typical microstructure of the central part of the weld.

**Figure 6 materials-15-09031-f006:**
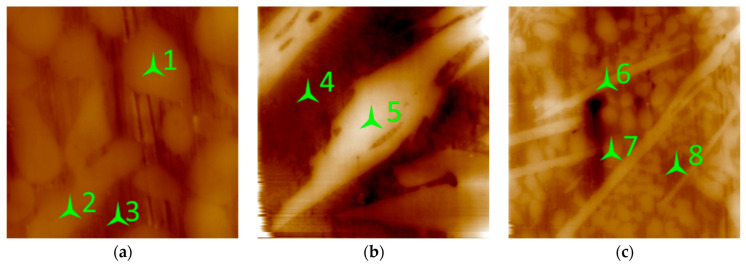
The AFM pattern and the location of measurement points: (**a**) small carbides in the center of weld 2, 15 × 15 µm, (**b**) the tip of long boride in the transition zone between welds 2 and 3, 40 × 40 µm, and (**c**) long borides and small carbides in the transition zone between welds 1 and 2, 40 × 40 µm.

**Figure 7 materials-15-09031-f007:**
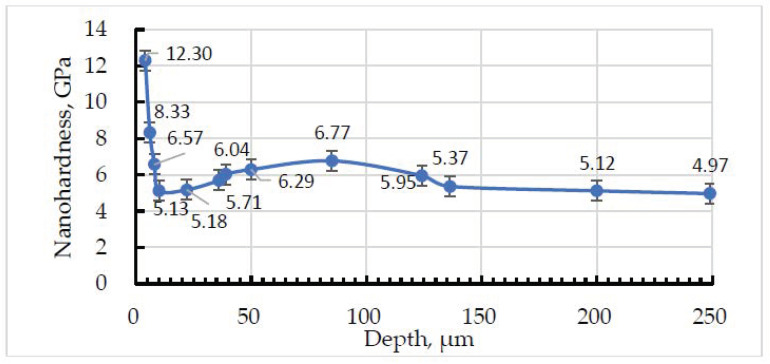
The in-depth hardness profile of matrix material below the friction surface of a sample tested at 15 MPa.

**Figure 8 materials-15-09031-f008:**
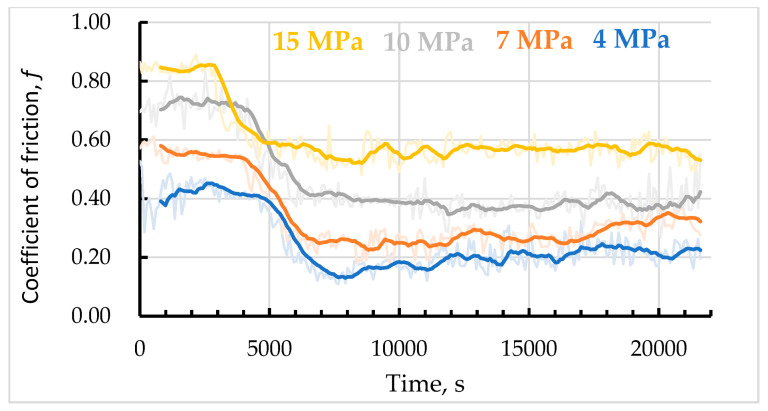
Coefficient of friction of the alloy samples tested at 4, 7, 10, and 15 MPa.

**Figure 9 materials-15-09031-f009:**
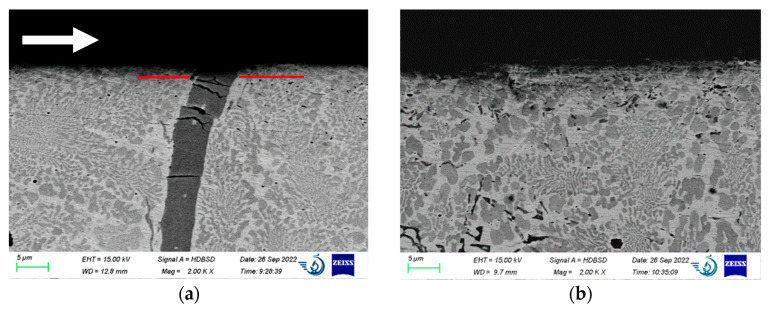
The microstructure (cross-section) of the alloy below the friction surface, 15 MPa. The arrow indicates the friction direction for both images: the bent and cracked boride grain (**a**) and uniform microstructure without coarse borides (**b**). White arrow shows the friction direction for both micrographs.

**Figure 10 materials-15-09031-f010:**
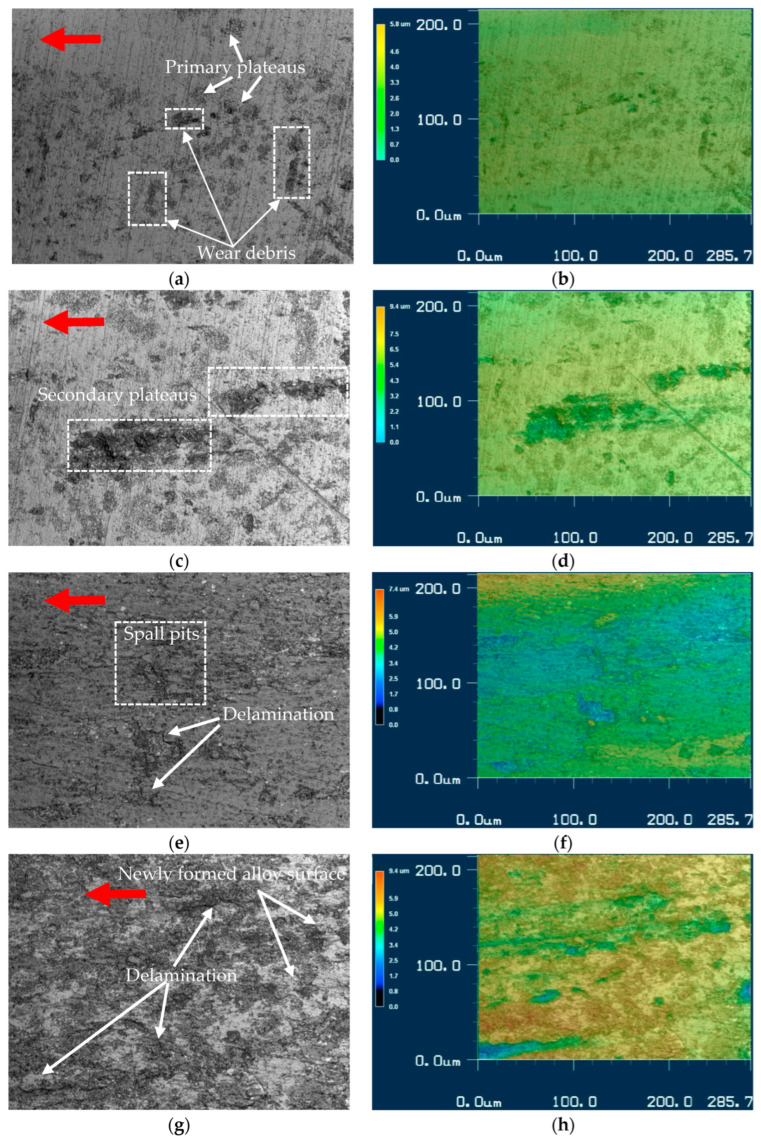
The friction surface topography (**a**,**c**,**e**,**g**) and 3D maps (**b**,**d**,**f**,**h**) of samples’ wear scars. (**a**,**b**): 4 MPa, Rz = 1.96 μm, Ra = 0.22 μm; (**c**,**d**): 7 MPa, Rz = 2.89 μm, Ra = 0.29 μm; (**e**,**f**): 10 MPa, Rz = 3.57 μm, Ra = 0.37 μm; and (**g**,**h**): 15 MPa, Rz = 6.1 μm, Ra = 0.56 μm.

**Figure 11 materials-15-09031-f011:**
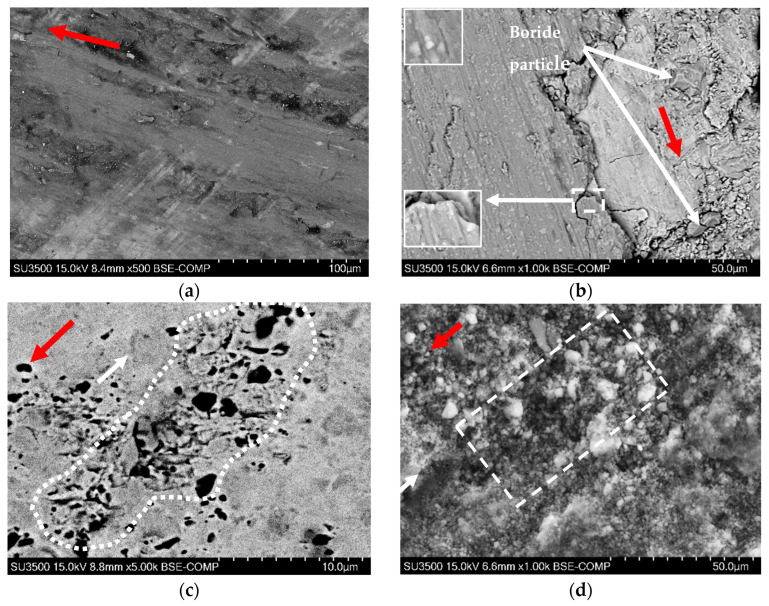
The micrographs of the friction surface of samples: (**a**) general surface, 4 MPa; (**b**) the microstructure of the secondary contact plateaus and apexes of vertically oriented boride grains (Figure 9a), 10 MPa, magnification of inserts, 5000; (**c**) top view of the rosette carbides on the friction surface, 7 MPa; and (**d**) wear debris on the friction surface, 15 MPa. Red arrows show the friction direction.

**Figure 12 materials-15-09031-f012:**
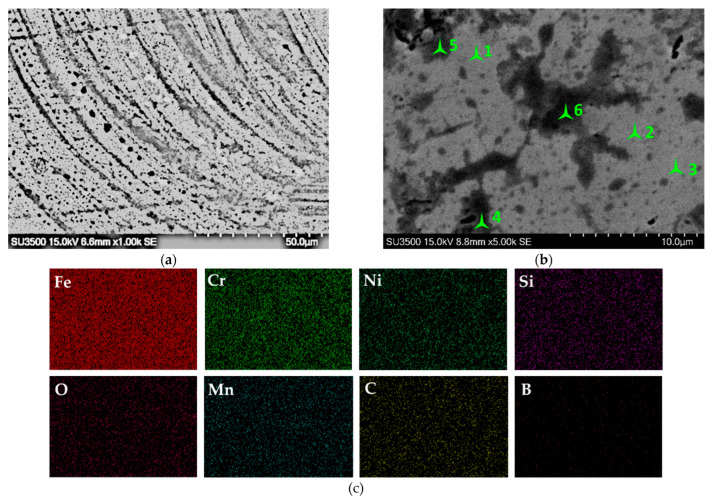
The scanned spectra positions (**a**); mapped general friction surface (**b**); and EDS mapping (**c**) of the general friction surface tested at 4 MPa.

**Figure 13 materials-15-09031-f013:**
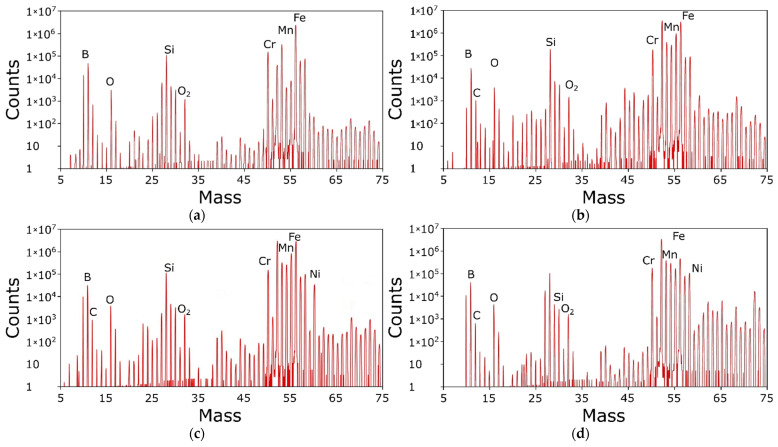
Mass spectra of the eutectic alloy coating surface at unit pressures of (**a**) 4 MPa, (**b**) 7 MPa, (**c**) 10 MPa, and (**d**) 15 MPa.

**Table 1 materials-15-09031-t001:** Composition of materials used for this research.

Type of Alloy	Chemical Composition, wt.%								
	Fe	C	Mn	B	Si	Cr	Ni	S	P
Filling powder	Bal.	0.5	12	3.3	4.4	12.0	11.6	0.01	0.03
Coating composition	Bal.	0.24	5.1	1.36	1.8	5	4.8	0.019	0.02
Steel 06JA	Bal.	0.06	0.3	–	0.02	0.1	0.1	0.025	0.02
Steel AISI 1045	Bal.	0.5	0.8	–	0.4	0.3	0.3	0.04	0.04

**Table 2 materials-15-09031-t002:** The results of EDS point analyses of the coating components.

Name	Spectrum, wt.%							
	1	2	3	4	5	6	7	8
C	23.58	22.11	5.86	4.9	19.86	19.67	6.08	5.63
Si	0.42	1.18	1.95	1.79	0.78	0.96	3.73	3.97
Cr	3.37	2.40	6.73	7.34	2.62	1.34	3.78	4.25
Mn	1.90	3.02	6.79	7.42	1.67	1.95	3.3	2.93
Fe	69.28	68.00	70.44	69.37	73.55	74.44	78.05	77.69
Ni	2.45	3.29	8.23	9.18	1.52	1.64	5.06	5.53
Total	100.00	100.00	100.00	100.00	100.00	100.00	100.00	100.00

**Table 3 materials-15-09031-t003:** The nanohardness values of the coating constituents.

Indent	Figure	Er, (GPa)	H, (GPa)	Image Size
1	Figure 6a	246.58	19.059	15 × 15 µm
2		248.95	19.782	
3		189.96	3.801	
4	Figure 6b	282.94	6.171	40 × 40 µm
5		305.66	23.447	
6	Figure 6c	277.9	22.893	40 × 40 µm
7		223.62	16.02	
8		195.13	4.586	

**Table 4 materials-15-09031-t004:** Values of the coefficient of friction measured during the wear test and wear loss.

№	Contact Pressure, MPa	Coefficient of Friction				Weight Wear Loss, mg
		Max	Average	Section 1	Section 2	
1	4	0.53	0.25	0.42	0.2	62
2	7	0.61	0.34	0.55	0.28	118
3	10	0.83	0.46	0.72	0.39	197
4	15	0.89	0.61	0.84	0.57	458

**Table 5 materials-15-09031-t005:** The composition of the spectra.

Element	Spectrum, wt.%						EDS (Figure 11c)
	1	2	3	4	5	6	
B	N/D	N/D	N/D	N/D	N/D	N/D	0.61
C	3.8	6.2	7.49	12.17	11.8	15.96	8.21
O	0.12	0.17	0.24	0.58	0.42	0.71	0.34
Si	2.36	1.36	1.02	1.32	1.64	0.99	1.29
Cr	9.11	12.59	8.32	6.13	8.23	4.91	11.18
Mn	2.25	0.0	1.78	1.99	2.52	1.49	2.06
Fe	65.33	66.41	60.79	59.54	66.58	52.37	61.41
Ni	16.92	13.23	8.23	18.28	14.73	23.58	14.9
Total	100.00	100.00	100.00	100.00	100.00	100.00	–

**Table 6 materials-15-09031-t006:** The SIMS data of the friction surface, wt.%.

Element	Contact Pressure, MPa				
	Intact	4 MPa	7 MPa	10 MPa	15 MPa
B	0.177	0.778	0.524	0.456	0.594
C	0.118	0.134	0.1398	0.15	0.1386
O	0.11	0.12	0.25	1.55	3.72
Si	2.23	2.119	2.629	1.978	1.754
Cr	11.68	12.98	14.286	16.71	18.07
Mn	9.69	8.805	9.455	8.286	3.017
Fe	59.31	60.66	59.746	58.969	62.868
Ni	16.47	14.25	13.76	13.22	13.32
Impurities	Bal.	Bal.	Bal.	Bal.	Bal.

## Data Availability

The data that support the findings of this study are available from the corresponding author upon reasonable request.

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
