# Peer review of "Microstructure and Friction Response of a Novel Eutectic Alloy Based on the Fe-C-Mn-B System"

_materials, 2022, doi:10.3390/ma15249031_

Round 1

Reviewer 1 Report

In chapter Materials- must include the mettallographic attack used for analysis.

line 178- "microporosities", instead porosities

line 204- after figure 3- X-rays diffraction, figure 4 follows not figure 2 

lines 229, 230, 231, 232, 233 are written with smaller letters, than the other text, please modify

Author Response

Dear Reviewer,

We appreciate the constructive suggestions enclosed in your review. All the authors of this manuscript are grateful for Reviewer’s constructive criticism that is intended to improve our work. We carefully considered the comments and herein we explain how we revised the manuscript based on your recommendations.

It is our belief that the manuscript is substantially improved after making the suggested edits. We want to express our appreciation for taking the time and effort necessary to provide guidance.

Yours sincerely,

Oleksandr Tisov

Xi’an Jiaotong University

China

Reviewer 2 Report

The authors present a manuscript about the tribological study of an Fe-C-Mn-B-Cr-Ni-Si alloy, which contains several measurements on the tested sample. However, the results are very useful, the described alloy is promising, I am not fully satisfyed with the manuscript. After fixing these issues, I recommend the manuscript for publication.

Because of to the various tests, the manuscript is tooo lengthy the reader can get lost in the details, maybe a few subsections of section 3 should be left out, or move to an appendix, to focus on the main results better. Most of the findings are discussed in section 3, the discussion (section 4) is  just the summary of the previous section.

In the manuscript, there are two Fig. 10s, and there are two Fig 11s. Figure 2 is written instead of Figure 4 at line 204. Please correct the numbering. Also, a general recommendation for the figures: use higher resulution, the texts on some of the figures look very bad. There are also misprints in the text, like 3 MPa is written in tale 6, instead of 4 MPa.

What was the surface temperature during the experiments? According to the data included in the text, during the 4 MPa experiment, about 700 kJ work was needed to move the sample by 8640 m, with the given friction coefficient, at 4 MPa pressure, which work would be enough to melt about 2.4 kg of iron. (At 15 MPa, the work would be enough to melt about 24 kg of iron.) Based on this simple estimation, I expect high surface temperatures due to the friction, which temperature can cause changes in the material, which would be present even without any friction, or wear. Please discuss this in the paper.

In real applications, different lubricants are used to decrease friction and cool the surface. These lubricants (and the much smaller rate of wear) can have large effects. What the authors think about this? Could we observe the same changes in the material in the presence of lubricants?

The friction coefficents decreases during the experiments, while the surface roughness increases. One can expect, that the friction coefficient is proportional to the surface roughness, but here we can observe the opposite. What is the explanation?

Author Response

(The authors gave the same response as above.)

Reviewer 3 Report

The authors of the paper “Microstructure and friction response of novel eutectic alloy based on Fe-C-Mn-B system” have investigated the microstructure and tribological properties of high alloyed steel. The authors have made good experimental work. The authors described the technology of the material production and investigated the microstructure of the coating. However, some points in the manuscript are questionable. The manuscript may be accepted for publication after revision accordingly following comments:

1.                  The title of the manuscript does not correspond to the content of the paper. Manganese is not the determining alloying element. Its concentration is the same as Si, Cr, and Ni. It is better to leave in the title the only eutectic components such as carbon and boron.

2.                  In the introduction part, the authors should describe a potential application of their materials.

3.                  The phase composition of the steel with high boron and Cr content should have a complex (Fe, Cr)2B phase. It is recommended to revise Figure 3. Please, consider the works of Pozdniakov et. Al for additional information.

4.                  Some peaks in Figure 3 are not recognized. It is recommended to analyze the phase composition more carefully.

5.                  It is hardly correct to use as a counterpart such a soft material as AISI 1045 steel. The hardness of the counterpart should have the same level as the hardness of the testing material.

6.                  Minor correction:

-          Standard deviations should be added to Figure 7 and the value of the concentrations in Tables 2-6.

-          Line 163: 10°/s should be changed to 10°/step to prevent misleading the readers. The scanning rate of 10°/second is too high.

Author Response

(The authors gave the same response as above.)

Round 2

Reviewer 3 Report

The authors have answered previous comments and improved the manuscript.  The paper may be accepted for publication.